# Masked World Models for Visual Control

Younggyo Seo[1,2,*]   Danijar Hafner[2,3,4]   Hao Liu[2]   Fangchen Liu[2]

Stephen James[2,†]   Kimin Lee[3]   Pieter Abbeel[2]

[1] KAIST   [2] UC Berkeley   [3] Google Research   [4] University of Toronto

**Abstract:** Visual model-based reinforcement learning (RL) has the potential to enable sample-efficient robot learning from visual observations. Yet the current approaches typically train a single model end-to-end for learning both visual representations and dynamics, making it difficult to accurately model the interaction between robots and small objects. In this work, we introduce a visual model-based RL framework that decouples visual representation learning and dynamics learning. Specifically, we train an autoencoder with convolutional layers and vision transformers (ViT) to reconstruct pixels given masked convolutional features, and learn a latent dynamics model that operates on the representations from the autoencoder. Moreover, to encode task-relevant information, we introduce an auxiliary reward prediction objective for the autoencoder. We continually update both autoencoder and dynamics model using online samples collected from environment interaction. We demonstrate that our decoupling approach achieves state-of-the-art performance on a variety of visual robotic tasks from Meta-world and RLBench, *e.g.*, we achieve 81.7% success rate on 50 visual robotic manipulation tasks from Meta-world, while the baseline achieves 67.9%. Code is available on the project website: https://sites.google.com/view/mwm-rl.

## 1   Introduction

Model-based reinforcement learning (RL) holds the promise of sample-efficient robot learning by learning a world model and leveraging it for planning [1, 2, 3] or generating imaginary states for behavior learning [4, 5]. These approaches have also previously been applied to environments with visual observations, by learning an action-conditional video prediction model [6, 7] or a latent dynamics model that predicts compact representations in an abstract latent space [8, 9]. However, learning world models on environments with complex visual observations, *e.g.,* accurately modeling interactions with small objects, is an open challenge.

We argue that this difficulty comes from the design of current approaches that typically optimize the world model end-to-end for learning both visual representations and dynamics [9, 10]. This imposes a trade-off between learning representations and dynamics that can prevent world models from accurately capturing visual details, making it difficult to predict forward into the future. Another approach is to learn representations and dynamics separately, such as earlier work by Ha and Schmidhuber [11] who train a variational autoencoder (VAE) [12] and a dynamics model on top of the VAE features. However, separately-trained VAE representations may not be amenable to dynamics learning [8, 10] or may not capture task-relevant details [11].

On the other hand, masked autoencoders (MAE) [13] have recently been proposed as an effective and scalable approach to visual representation learning, by training a self-supervised vision transformer (ViT) [14] to reconstruct masked patches. While it motivates us to learn world models on top of MAE representations, we find that MAE often struggles to capture fine-grained details within patches. Because capturing visual details, *e.g.,* object positions, is crucial for solving visual control tasks, it is desirable to develop a representation learning method that captures such details but also achieves the benefits of MAE such as stability, compute-efficiency, and scalability.

---

[*]Work done while visiting UC Berkeley. Correspondence to younggyo.seo@kaist.ac.kr.

[†]Now at Dyson Robot Learning Lab.

6th Conference on Robot Learning (CoRL 2022), Auckland, New Zealand.

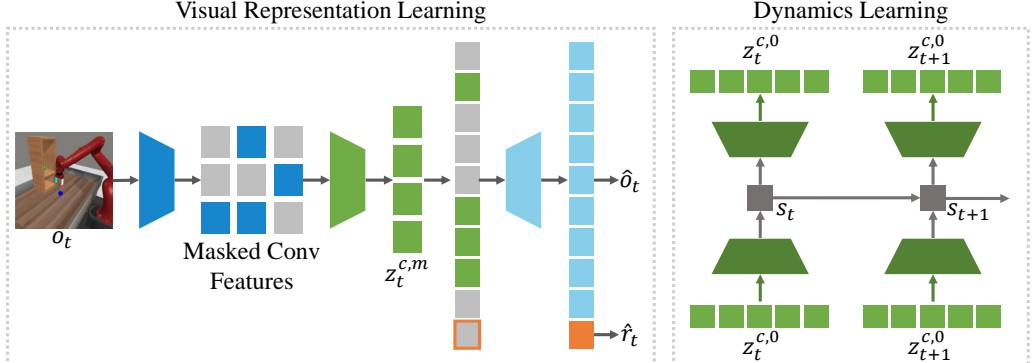

Figure 1: Illustration of our approach. We continually update visual representations and dynamics using online samples collected from environment interaction, by repeating iterative processes of training (Left) an autoencoder with convolutional feature masking and reward prediction and (Right) a latent dynamics model in the latent space of the autoencoder. We note that autoencoder parameters are not updated during dynamics learning.

In this paper, we present Masked World Models (MWM), a visual model-based RL algorithm that decouples visual representation learning and dynamics learning. The key idea of MWM is to train an autoencoder that reconstructs visual observations with convolutional feature masking, and a latent dynamics model on top of the autoencoder. By introducing early convolutional layers and masking out convolutional features instead of pixel patches, our approach enables the world model to capture fine-grained visual details from complex visual observations. Moreover, in order to learn task-relevant information that might not be captured solely by the reconstruction objective, we introduce an auxiliary reward prediction task for the autoencoder. Specifically, we separately update visual representations and dynamics by repeating the iterative processes of (i) training the autoencoder with convolutional feature masking and reward prediction, and (ii) learning the latent dynamics model that predicts visual representations from the autoencoder (see Figure 1).

**Contributions** We highlight the contributions of our paper below:

- We demonstrate the effectiveness of decoupling visual representation learning and dynamics learning for visual model-based RL. MWM significantly outperforms a state-of-the-art model-based baseline [15] on various visual control tasks from Meta-world [16] and RLBench [17].

- We show that a self-supervised ViT trained to reconstruct visual observations with convolutional feature masking can be effective for visual model-based RL. Interestingly, we find that masking convolutional features can be more effective than pixel patch masking [13], by allowing for capturing fine-grained details within patches. This is in contrast to the observation in Touvron et al. [18], where both perform similarly on the ImageNet classification task [19].

- We show that an auxiliary reward prediction task can significantly improve performance by encoding task-relevant information into visual representations.

## 2 Related Work

**World models from visual observations** There have been several approaches to learn visual representations for model-based approaches via image reconstruction [6, 7, 8, 9, 10, 11, 15, 20, 21, 22], *e.g.,* learning a video prediction model [6, 23] or a latent dynamics model [8, 9, 10]. This has been followed by a series of works that demonstrated the effectiveness of model-based approaches for solving video games [15, 24, 22] and visual robot control tasks [7, 21, 25, 26]. There also have been several works that considered different objectives, including bisimulation [27] and contrastive learning [28, 29, 30]. While most prior works optimize a single model to learn both visual representations and dynamics, we instead develop a framework that decouples visual representation learning and dynamics learning.

**Self-supervised vision transformers** Self-supervised learning with vision transformers (ViT) [14] has been actively studied. For instance, Chen et al. [31] introduced MoCo-v3 which trains a ViT with

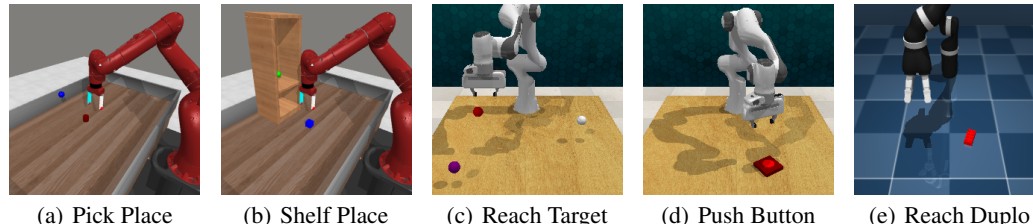

| (a) Pick Place | (b) Shelf Place | (c) Reach Target | (d) Push Button | (e) Reach Duplo |

Figure 2: Examples of visual observations used in our experiments. We consider a variety of visual robot control tasks from Meta-world [16], RLBench [17], and DeepMind Control Suite [41].

contrastive learning. Caron et al. [32] introduced DINO which utilizes a self-distillation loss [33], and demonstrated that self-supervised ViTs contain information about the semantic layout of images. Training self-supervised ViTs with masked image modeling [13, 34, 35, 36, 37, 38, 39] has also been successful. In particular, He et al. [13] proposed a masked autoencoder (MAE) that reconstructs masked pixel patches with an asymmetric encoder-decoder architecture. Unlike MAE, we propose to randomly mask features from early convolutional layers [40] instead of pixel patches and demonstrate that self-supervised ViTs can also be effective for visual model-based RL.

We provide more discussion on related works in more detail in Appendix C.

## 3 Preliminaries

**Problem formulation**  We formulate a visual control task as a partially observable Markov decision process (POMDP) [42], which is defined as a tuple $(\mathcal{O}, \mathcal{A}, p, r, \gamma)$. $\mathcal{O}$ is the observation space, $\mathcal{A}$ is the action space, $p\left(o_t | o_{<t}, a_{<t}\right)$ is the transition dynamics, $r$ is the reward function that maps previous observations and actions to a reward $r_t = r\left(o_{\leq t}, a_{<t}\right)$, and $\gamma \in [0, 1)$ is the discount factor.

**Dreamer**  Dreamer [15, 21] is a visual model-based RL method that learns world models from pixels and trains an actor-critic model via latent imagination. Specifically, Dreamer learns a Recurrent State Space Model (RSSM) [9], which consists of following four components:

$$
\begin{array}{llll}
\text{Representation model:} & s_t \sim q_\theta(s_t \,|\, s_{t-1}, a_{t-1}, o_t) & \text{Image decoder:} & \hat{o}_t \sim p_\theta(\hat{o}_t \,|\, s_t) \\
\text{Transition model:} & \hat{s}_t \sim p_\theta(\hat{s}_t \,|\, s_{t-1}, a_{t-1}) & \text{Reward predictor:} & \hat{r}_t \sim p_\theta(\hat{r}_t \,|\, s_t)
\end{array} \tag{1}
$$

The representation model extracts model state $s_t$ from previous model state $s_{t-1}$, previous action $a_{t-1}$, and current observation $o_t$. The transition model predicts future state $\hat{s}_t$ without the access to current observation $o_t$. The image decoder reconstructs raw pixels to provide learning signal, and the reward predictor enables us to compute rewards from future model states without decoding future frames. All model parameters $\theta$ are trained to jointly learn visual representations and environment dynamics by minimizing the negative variational lower bound [12]:

$$
\begin{aligned}
\mathcal{L}(\theta) \doteq \mathbb{E}_{q_\theta(s_{1:T}|a_{1:T}, o_{1:T})} \Big[ & \\
\textstyle\sum_{t=1}^{T} \Big( & -\ln p_\theta(o_t|s_t) - \ln p_\theta(r_t|s_t) + \beta\, \mathrm{KL}\left[q_\theta(s_t|s_{t-1}, a_{t-1}, o_t) \,\|\, p_\theta(\hat{s}_t|s_{t-1}, a_{t-1})\right] \Big) \Big],
\end{aligned} \tag{2}
$$

where $\beta$ is a hyperparameter that controls the tradeoff between the quality of visual representation learning and the accuracy of dynamics learning [43]. Then, the critic is learned to regress the values computed from imaginary rollouts, and the actor is trained to maximize the values by propagating analytic gradients back through the transition model (see Appendix A for the details).

**Masked autoencoder**  Masked autoencoder (MAE) [13] is a self-supervised visual representation technique that trains an autoencoder to reconstruct raw pixels with randomly masked patches consisting of pixels. Following a scheme introduced in vision transformer (ViT) [14], the observation $o_t \in \mathbb{R}^{H \times W \times C}$ is processed with a patchify stem that reshapes $o_t$ into a sequence of 2D patches $h_t \in \mathbb{R}^{N \times (P^2 C)}$, where $P$ is the patch size and $N = HW/P^2$ is the number of patches. Then a subset of patches is randomly masked with a ratio of $m$ to construct $h_t^m \in \mathbb{R}^{M \times (P^2 C)}$.

$$
\text{Patchify stem:} \quad h_t = f_\phi^{\texttt{patch}}(o_t) \qquad \text{Masking:} \quad h_t^m \sim p^{\texttt{mask}}(h_t^m \,|\, h_t, m) \tag{3}
$$

A ViT encoder embeds only the remaining patches $h_t^m$ into $D$-dimensional vectors, concatenates the embedded tokens with a learnable CLS token, and processes them through a series of Transformer layers [44]. Finally, a ViT decoder reconstructs the observation by processing tokens from the encoder and learnable mask tokens through Transformer layers followed by a linear output head:

$$\text{ViT encoder:} \quad z_t^m \sim p_\phi(z_t^m \mid h_t^m) \quad \text{ViT decoder:} \quad \hat{o}_t \sim p_\phi(\hat{o}_t \mid z_t^m) \tag{4}$$

All the components paramaterized by $\phi$ are jointly optimized to minimize the mean squared error (MSE) between the reconstructed and original pixel patches. MAE computes $z_t^0$ without masking, and utilizes its first component (*i.e.,* CLS representation) for downstream tasks (*e.g.,* image classification).

# 4   Masked World Models

In this section, we present Masked World Models (MWM), a visual model-based RL framework for learning accurate world models by separately learning visual representations and environment dynamics. Our method repeats (i) updating an autoencoder with convolutional feature masking and an auxiliary reward prediction task (see Section 4.1), (ii) learning a dynamics model in the latent space of the autoencoder (see Section 4.2), and (iii) collecting samples from environment interaction. We provide the overview and pseudocode of MWM in Figure 1 and Appendix D, respectively.

## 4.1   Visual Representation Learning

It has been observed that masked image modeling with a ViT architecture [13, 34, 36] enables compute-efficient and stable self-supervised visual representation learning. This motivates us to adopt this approach for visual model-based RL, but we find that masked image modeling with commonly used pixel patch masking [13] often makes it difficult to learn fine-grained details within patches, *e.g.,* small objects (see Appendix B for a motivating example). While one can consider small-size patches, this would increase computational costs due to the quadratic complexity of self-attention layers.

To handle this issue, we instead propose to train an autoencoder that reconstructs raw pixels given randomly masked convolutional features. Unlike previous approaches that utilize a patchify stem and randomly mask pixel patches (see Section 3), we adopt a convolution stem [14, 40] that processes $o_t$ through a series of convolutional layers followed by a flatten layer, to obtain $h_t^c \in \mathbb{R}^{N_c \times D}$ where $N_c$ is the number of convolutional features. Then $h_t^c$ is randomly masked with a ratio of $m$ to obtain $h_t^{c,m} \in \mathbb{R}^{M_c \times D}$, and ViT encoder and decoder process $h_t^{c,m}$ to reconstruct raw pixels.

$$\begin{array}{llll} \text{Convolution stem:} & h_t^c = f_\phi^{\texttt{conv}}(o_t) & \text{Masking:} & h_t^{c,m} \sim p^{\texttt{mask}}(h_t^{c,m} \mid h_t^c, m) \\ \text{ViT encoder:} & z_t^{c,m} \sim p_\phi(z_t^{c,m} \mid h_t^{c,m}) & \text{ViT decoder:} & \hat{o}_t \sim p_\phi(\hat{o}_t \mid z_t^{c,m}) \end{array} \tag{5}$$

Because early convolutional layers mix low-level details, we find that our autoencoder can effectively reconstruct all the details within patches by learning to extract information from nearby non-masked features (see Figure 7 for examples). This enables us to learn visual representations capturing such details while also achieving the benefits of MAE, *e.g.,* stability and compute-efficiency[3].

**Reward prediction**  In order to encode task-relevant information that might not be captured solely by the reconstruction objective, we introduce an auxiliary objective for the autoencoder to predict rewards jointly with pixels. Specifically, we make the autoencoder predict the reward $r_t$ from $z_t^{c,m}$ in conjunction with raw pixels.

$$\text{ViT decoder with reward prediction:} \quad \hat{o}_t, \hat{r}_t \sim p_\phi(\hat{o}_t, \hat{r}_t \mid z_t^{c,m}) \tag{6}$$

In practice, we concatenate one additional learnable mask token to inputs of the ViT decoder, and utilize the corresponding output representation for predicting the reward with a linear output head.

**High masking ratio**  Introducing early convolutional layers might impede the masked reconstruction tasks because they propagate information across patches [18], and the model can exploit this to find a shortcut to solve reconstruction tasks. However, we find that a high masking ratio (*i.e.,* 75%) can prevent the model from finding such shortcuts and induce useful representations (see Figure 6(b) for supporting experimental results). This also aligns with the observation from Touvron et al. [18], where masked image modeling [34] with a convolution stem [45] can achieve competitive performance with the patchify stem on the ImageNet classification task [19].

---

[3]We also note that Xiao et al. [40] showed that introducing early convolutional layers can stabilize the training of ViT models, which can be helpful for RL that requires stability from the beginning of the training.

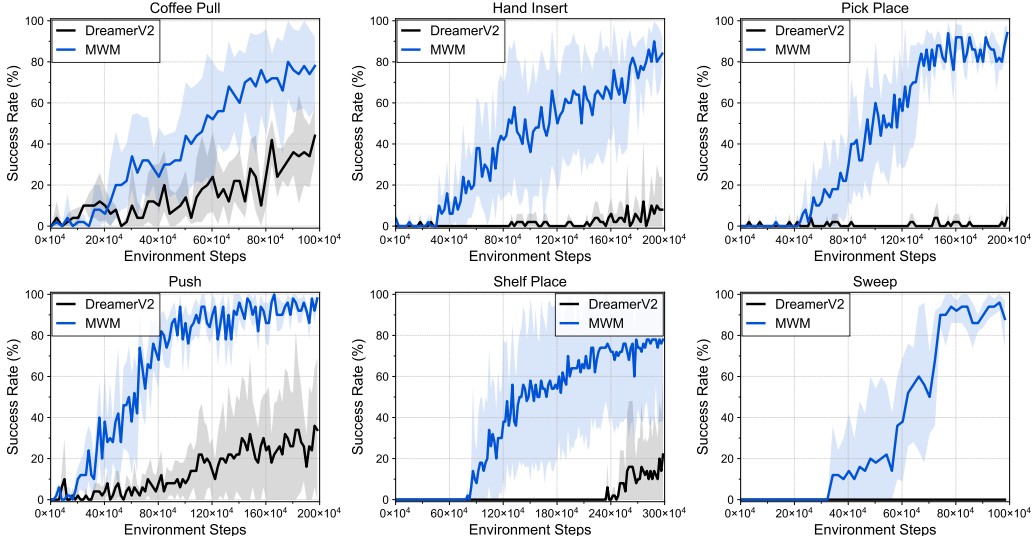

Figure 3: Learning curves on six visual robotic manipulation tasks from Meta-world as measured on the success rate. We select the tasks that require modeling interactions between small objects and robot arms. Learning curves on 50 tasks are available in Appendix G. The solid line and shaded regions represent the mean and bootstrap confidence intervals, respectively, across five runs.

## 4.2   Latent Dynamics Learning

Once we learn visual representations, we leverage them for efficiently learning a dynamics model in the latent space of the autoencoder. Specifically, we obtain the frozen representations $z_t^{c,0}$ from the autoencoder, and then train a variant of RSSM whose inputs and reconstruction targets are $z_{t,0}^c$, by replacing the representation model and the image decoder in Equation 1 with following components:

$$\begin{aligned}
\text{Representation model:} &\quad s_t \sim q_\theta(s_t \,|\, s_{t-1}, a_{t-1}, z_t^{c,0}) \\
\text{Visual representation decoder:} &\quad \hat{z}_t^{c,0} \sim p_\theta(\hat{z}_t^{c,0} \,|\, s_t)
\end{aligned} \tag{7}$$

Because visual representations capture both high- and low-level information in an abstract form, the model can focus more on dynamics learning by reconstructing them instead of raw pixels (see Section 5.5 for relevant discussion). Here, we also note that we utilize all the elements of $z_t^{c,0}$ unlike MAE that only utilizes CLS representation for downstream tasks. We empirically find this enables the model to receive rich learning signals from reconstructing all the representations containing spatial information (see Appendix I for supporting experiments).

**Optimization**   Given a random batch $\{(o_j, r_j, a_j)\}_{j=1}^B$, MWM objective is defined as follows:

$$\mathcal{L}^{\texttt{mwm}}(\phi, \theta) = \frac{1}{B} \sum_{j=1}^B \Big( \underbrace{-\ln p_\phi(o_j | z_j^{c,m}) - \ln p_\phi(r_j | z_j^{c,m})}_{\text{visual representation learning}} \tag{8}$$

$$\underbrace{- \ln p_\theta(z_j^{c,0} | s_j) - \ln p_\theta(r_j | s_j) + \beta \, \text{KL}\left[q_\theta(s_j | s_{j-1}, a_{j-1}, z_j^{c,0}) \,\|\, p_\theta(\hat{s}_j | s_{j-1}, a_{j-1})\right]}_{\text{dynamics learning}} \Big)$$

## 5   Experiments

We evaluate MWM on various robotics benchmarks, including Meta-world [16] (see Section 5.1), RLBench [17] (see Section 5.2), and DeepMind Control Suite [46] (see Section 5.3). We remark that these benchmarks consist of diverse and challenging visual robotic tasks. We also analyze algorithmic design choices in-depth (see Section 5.4) and provide a qualitative analysis of how our decoupling approach works by visualizing the predictions from the latent dynamics model (see Section 5.5).

**Implementation**   We use visual observations of $64 \times 64 \times 3$. For the convolution stem, we stack 3 convolution layers with the kernel size of 4 and stride 2, followed by a linear projection layer.

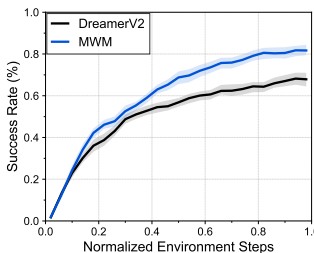 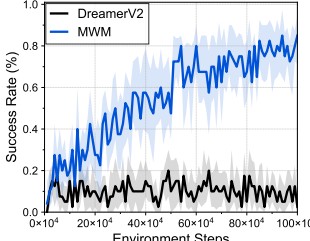 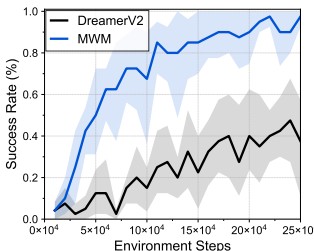

(a) Meta-world aggregated    (b) RLBench: Reach Target    (c) RLBench: Push Button

Figure 4: (a) Aggregate performance on all 50 Meta-world tasks. We normalize environment steps by maximum steps in each task. The solid line and shaded regions represent the mean and stratified bootstrap confidence intervals, respectively, across 250 runs. We report the learning curves on (b) Reach Target and (c) Push Button from RLBench. Performances are not directly comparable to previous results [47, 48] due to the difference in setups (see Section 5.2). The solid line and shaded regions represent the mean and bootstrap confidence intervals, respectively, across eight runs.

We use a 4-layer ViT encoder and a 3-layer ViT decoder. We find that initializing the autoencoder with a warm-up schedule at the beginning of training is helpful. Unlike MAE, we compute the loss on entire pixels because we do not apply masking to pixels. For world models, we build our implementation on top of DreamerV2 [15]. To take a sequence of autoencoder representations as inputs, we replace a CNN encoder and decoder with a 2-layer Transformer encoder and decoder. We use same hyperparameters within the same benchmark. More details are available in Appendix E.

## 5.1   Meta-world Experiments

**Environment details**   In order to use a single camera viewpoint consistently over all 50 tasks, we use the modified `corner2` camera viewpoint for all tasks. In our experiments, we classify 50 tasks into `easy`, `medium`, `hard`, and `very hard` tasks where experiments are run over 500K, 1M, 2M, 3M environments steps with action repeat of 2, respectively. More details are available in Appendix F.

**Results**   In Figure 3, we report the performance on a set of selected six challenging tasks that require agents to control robot arms to interact with small objects. We find that MWM significantly outperforms DreamerV2 in terms of both sample-efficiency and final performance. In particular, MWM achieves $> 80\%$ success rate on Pick Place while DreamerV2 struggles to solve the task. These results show that our approach of separating visual representation learning and dynamics learning can learn accurate world models on challenging domains. We also note that DreamerV2 learns visual representations with the reward prediction loss, which implies that our superior performance does not come from utilizing additional information. Figure 4(a) shows the aggregate performance over all the 50 tasks from the benchmark, demonstrating that our method consistently outperforms DreamerV2 overall. We also provide learning curves on all individual tasks in Appendix G, where MWM consistently achieves similar or better performance on most tasks.

## 5.2   RLBench Experiments

**Environment details**   In order to evaluate our method on more challenging visual robotic manipulation tasks, we consider RLBench [17], which has previously acted as an effective proxy for real-robot performance [48]. Since RLBench consists of sparse-reward and challenging tasks, solving them typically requires expert demonstrations, specialized network architectures, additional inputs (e.g., point cloud and proprioceptive states), and an action mode that requires path planning [47, 48, 49, 50]. While we could utilize some of these components, we instead leave this as future work in order to maintain a consistent evaluation setup across multiple domains. In our experiments, we instead consider two relatively easy tasks with dense rewards, and utilize an action mode that specifies the delta of joint positions. We provide more details in Appendix F.

**Results**   As shown in Figure 4(b) and Figure 4(c), we observe that our approach can also be effective on RLBench tasks, significantly outperforming DreamerV2. In particular, DreamerV2

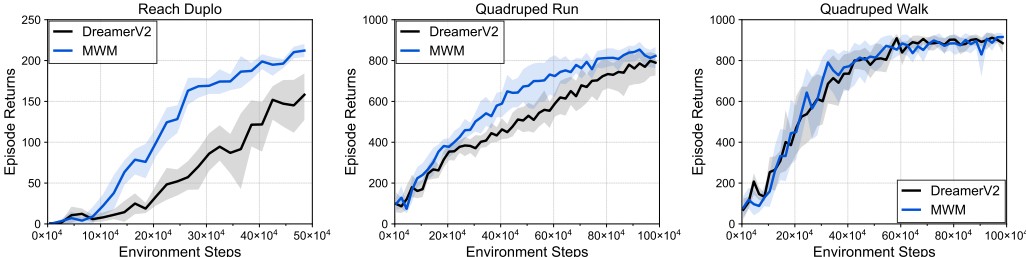

Figure 5: Learning curves on three visual robot control tasks from DeepMind Control Suite as measured on the episode return. The solid line and shaded regions represent the mean and bootstrap confidence intervals, respectively, across eight runs.

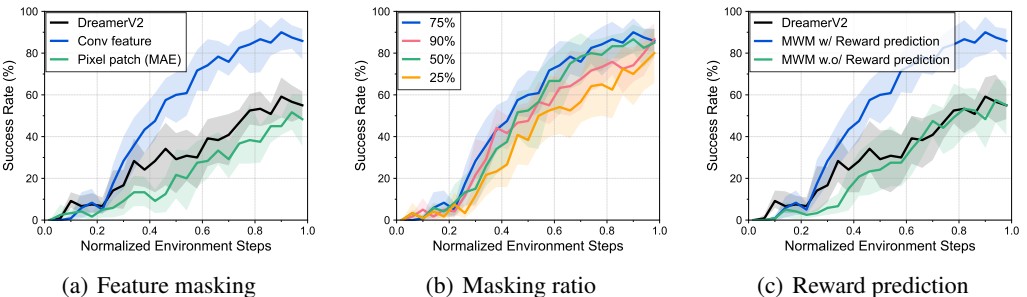

(a) Feature masking        (b) Masking ratio        (c) Reward prediction

Figure 6: Learning curves on three manipulation tasks from Meta-world that investigate the effect of (a) convolutional feature masking, (b) masking ratio, and (c) reward prediction. The solid line and shaded regions represent the mean and stratified bootstrap confidence interval across 12 runs.

achieves $< 20\%$ success rate on Reach Target, while our approach can solve the tasks with $> 80\%$ success rates. We find that this is because DreamerV2 fails to capture target positions in visual observations, while our method can capture such details (see Section 5.5 for relevant discussion and visualizations). However, we also note that these results are preliminary because they are still too sample-inefficient to be used for real-world scenarios. We provide more discussion in Section 6.

## 5.3 DeepMind Control Suite Experiments

**Environment details** In order to demonstrate that our approach is generally applicable to diverse visual control tasks, we also evaluate our method on visual locomotion tasks from the widely used DeepMind Control Suite benchmark. Following a standard setup in Hafner et al. [21], we use an action repeat of 2 and default camera configurations. We provide more details in Appendix F.

**Results** Figure 5 shows that our method achieves competitive performance to DreamerV2 on visual locomotion tasks (i.e., Quadruped tasks), demonstrating the generality of our approach across diverse visual control tasks. We also observe that our method outperforms DreamerV2 on Reach Duplo, which is one of a few manipulation tasks in the benchmark (see Figure 2(e) for an example). This implies that our method is effective on environments where the model should capture fine-grained details like object positions. More results are available in Appendix H, where trends are similar.

## 5.4 Ablation Study

**Convolutional feature masking** We compare convolutional feature masking with pixel masking (*i.e.,* MAE) in Figure 6(a), which shows that convolutional feature masking significantly outperforms pixel masking. Both results are achieved with the reward prediction. This demonstrates that enabling the model to capture fine-grained details within patches can be important for visual control. We also report the performance with varying masking ratio $m \in \{0.25, 0.5, 0.75, 0.9\}$ in Figure 6(b). As we discussed in Section 4.1, we find that $m = 0.75$ achieves better performance than $m \in \{0.25, 0.5\}$ because strong regularization can prevent the model from finding a shortcut from input pixels. However, we also find that too strong regularization (*i.e.,* $m = 0.9$) degrades the performance.

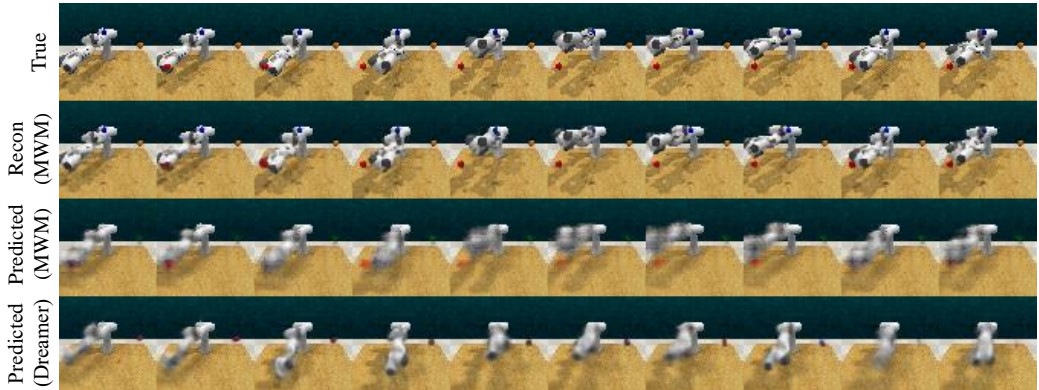

Figure 7: Future frames reconstructed with the autoencoder (*i.e.,* Recon) and predicted by latent dynamics models (*i.e.,* Predicted). Best viewed as video provided in Appendix B.

**Reward prediction**   In Figure 6(c), we find that performance significantly degrades without reward prediction, which shows that the reconstruction objective might not be sufficient for learning task-relevant information. It would be an interesting future direction to develop a representation learning scheme that learns task-relevant information without rewards because they might not be available in practice. We provide more ablation studies and learning curves on individual tasks in Appendix I. Moreover, we also show that reward prediction allows for learning useful representations that can be generally applicable to diverse robotic manipulation tasks in Appendix L.

## 5.5   Qualitative Analysis

We visually investigate how our world model works compared to the world model of DreamerV2. Specifically, we visualize the future frames predicted by latent dynamics models on Reach Target from RLBench in Figure 7. In this task, a robot arm should reach a target position specified by a red block in visual observations (see Figure 2(c)), which changes every trial. Thus it is crucial for the model to accurately predict the position of red blocks for solving the tasks. We find that our world model effectively captures the position of red blocks, while DreamerV2 fails. Interestingly, we also observe that our latent dynamics model ignores the components that are not task-relevant such as blue and orange blocks, though the reconstructions from the autoencoder are capturing all the details. This shows how our decoupling approach works: it encourages the autoencoder to focus on learning representations capturing the details and the dynamics model to focus on modeling task-relevant components of environments. We provide more examples in Appendix B.

## 6   Discussion

We have presented Masked World Models (MWM), which is a visual model-based RL framework that decouples visual representation learning and dynamics learning. By learning a latent dynamics model operating in the latent space of a self-supervised ViT, we find that our approach allows for solving a variety of visual control tasks from Meta-world, RLBench, and DeepMind Control Suite.

**Limitation**   Despite the results, there are a number of areas for improvement. As we have shown in Figure 6(c), the performance of our approach heavily depends on the auxiliary reward prediction task. This might be because our autoencoder is not learning temporal information, which is crucial for learning task-relevant information. It would be interesting to investigate the performance of video representation learning with ViTs [36, 51]. It would also be interesting to study introducing auxiliary prediction for other modalities, such as audio. Another weakness is that our model operates only on RGB pixels from a single camera viewpoint; we look forward to a future work that incorporates different input modalities such as proprioceptive states and point clouds, building on top of the recent multi-modal learning approaches [52, 53]. Finally, our approach trains behaviors from scratch, which makes it still too sample-inefficient to be used in real-world scenarios. Leveraging a small number of demonstrations, incorporating the action mode with path planning [47], or pre-training a world model on video datasets [54] are directions we hope to investigate in future works.

**Acknowledgments**

We would like to thank Jongjin Park and Sihyun Yu for helpful discussions. We also thank Cirrascale Cloud Services[4] for providing compute resources. This work was partially supported by Office of Naval Research under grant N00014-21-1-2769 and grant N00014-22-1-2121, the Darpa RACER program, the Hong Kong Centre for Logistics Robotics, BMW, and Institute of Information & Communications Technology Planning & Evaluation (IITP) grant funded by the Korea government (MSIT) (No.2019-0-00075, Artificial Intelligence Graduate School Program (KAIST)).

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
