# OpenReview forum: "Masked World Models for Visual Control"
_robot-learning.org/CoRL/2022/Conference — CoRL 2022 Poster_

### Official Review · Reviewer_GUgv · 2022-07-28

**Originality:** Good
**Technical Quality:** Good
**Clarity Of Presentation:** Very Good
**Impact:** 3

**Recommendation:**

Weak Accept: I recommend accepting the paper, but will not argue for my recommendation if the majority of other reviewers have a different opinion.

**Summary:**

This paper successfully achieved visual representation learning for world models in visual model-based RL with good accuracy using appropriate transformer models and masking processes.
As a result, the proposed method achieved higher success rates than conventional methods in variety of robot tasks.

**Issues:**

I think, for the versatility, it would be better to learn visual representation and dynamics together because they are task-irrevant (and ultimately, environment-irrevant) components, and the reward prediction should be decoupled from them.
It seems to me that some other task-irrelevant information should be utilized to prioritize observations, as discussed in Limitation.

The addition of the reward prediction finally led to exceed the baseline, so I feel that it would be more important than masking and decoupling, which are in the title of this paper.

I know the page limitation is strict, but the authors putted too much information in Appendix, causing bad readability.
At least, in my opinion, the optimization problem for the proposed method should be inserted in the main text, rather than the definition of Dreamer.

**Quality Of The Limitations Section:**

Additional details required

**Reviewer Expertise:**

3: The reviewer is fairly confident that the evaluation is correct

**Robotics Focus:**

Highly relevant to robotics but no hardware experiments

**Strengths And Weaknesses:**

Strengths:
- The proposed method was validated in a wide variety of ways and demonstrated sufficient experimental value.
- The intention of the model design was clearly explained qualitatively.

Weaknesses:
- Although the proposal is a decoupling of representation learning and dynamics learning, the reward function is included in representation learning, which makes the visual representation task-relevant and lacks versatility.
- Although the masking method was changed, its benefit was only shown qualitatively and experimentally, so I cannot find mathematical explanations.
- The analysis of why the learning curve was better than the baseline was insufficient, since only the learning curve was included in this paper.

**Summary Of Recommendation:**

The performance of the proposed method was experimentally verified by a large number of simulations, including ablation studies.
On the other hand, the proposed method was not mathematically convincing.
Too much emphasis was placed on exceeding the baseline in terms of the control performance, so further analysis may be needed to understand why the proposed method was better.

---

> ### Author Response · Authors · 2022-08-24
> **Response to Reviewer GUgv (1/2)**
>
> Dear reviewer GUgv,
>
> We sincerely appreciate your valuable comments. We found them extremely helpful in improving our draft. We address each comment in detail, one by one below.
>
> ---
>
> **Q1: Concern on the generalizability of learned representations due to reward prediction**
>
> **A1.** We agree that representation learning only with task-irrelevant information would be an interesting direction. Nevertheless, we remark that reward prediction on a specific task can also encourage visual representations to capture task-irrelevant information useful for solving various manipulation tasks. For instance, rewards are often designed to contain information about robot arm movements (translations and rotations) and fine-grained details about objects, which can be shared across various manipulation tasks.
>
> To empirically support this, we provide additional experimental results (https://imgur.com/a/IHuRxhZ) where we utilize frozen representations trained on push task for solving manipulation tasks with a difference to push task: (i) Push Back that requires moving block to a different direction, (ii) Pick Place that requires picking up the block, and (iii) Drawer Open that contains unseen drawer object in the observation. We observe that performance with frozen representations can be similar to or better than the performance of MWM trained from scratch, which shows that representations learned with reward prediction can be versatile.
>
> ---
>
> **Q2: Importance of masking and decoupling compared to reward prediction**
>
> **A2.** We would like to emphasize that the baseline is also learning encoders with gradients from the reward prediction. This shows that masking and decoupling are crucial factors for performance improvement over the baseline from our approach. We will clarify this.
>
> ---
>
> **Q3: Mathematical explanation on the benefit from the changed masking method**
>
> **A3.** Because masking out convolutional features can be seen as applying dropout to features from early convolution layers, obtaining autoencoder representations without masking can be interpreted as averaging the representations from sub-networks with masked features [1, 2, 3], which could explain the benefit from our masking approach. Moreover, while not being a mathematical explanation, we would like to refer the reviewer to [4], which demonstrates that introducing early convolution layers can stabilize and accelerate the training process of the ViT model. This could be especially important for RL, where the training can be unstable especially at the beginning of the training. We will include the relevant discussion in the final draft.
>
> [1] Srivastava, Nitish, Geoffrey Hinton, Alex Krizhevsky, Ilya Sutskever, and Ruslan Salakhutdinov. "Dropout: a simple way to prevent neural networks from overfitting." The Journal of Machine Learning Research, 2014
>
> [2] Baldi, Pierre, and Peter J. Sadowski. "Understanding dropout." Advances in Neural Information Processing Systems, 2013.
>
> [3] Gal, Yarin, and Zoubin Ghahramani. "Dropout as a bayesian approximation: Representing model uncertainty in deep learning." International Conference on Machine Learning, 2016.
>
> [4] Xiao, Tete, Mannat Singh, Eric Mintun, Trevor Darrell, Piotr Dollár, and Ross Girshick. "Early convolutions help transformers see better." Advances in Neural Information Processing Systems, 2021.

---

> > ### Author Response · Authors · 2022-08-24
> > **Response to Reviewer GUgv (2/2)**
> >
> > **Q4: Concern on insufficient analysis to understand how MWM works**
> >
> > **A4.** Thanks for the comment. We would like to remark that we have provided a qualitative analysis of how MWM can be better for capturing task-relevant information than DreamerV2 in Figure 7 and how masked autoencoder without early convolution could fail to capture fine-grained details in Figure 8 of the Appendix. To further address your concern, we additionally provide an analysis that investigates how the reward prediction enables the model to achieve better performance. Here, our hypothesis is that reward prediction can provide information useful for robotic manipulation, which can also be shared across various tasks. For instance, rewards are often designed to contain information about robot arm movements (translations and rotations) and fine-grained details about objects, which can be shared across various manipulation tasks.
> >
> > To empirically investigate this hypothesis, we first train MWM on Meta-world Push task with and without reward prediction, and train a regression model to predict the states in Push (seen) and Pick-place (unseen) tasks on top of frozen autoencoder representations from Push task. In our experiments (https://imgur.com/a/2rnTBje), we observe that representations trained with reward prediction consistently achieve low prediction error. Moreover, we also report the performance of MWM on Pick-place with these frozen representations, where we find that reward prediction enables generalization to the significantly different Pick-place task. This supports that reward prediction can indeed provide useful information for robotic manipulation.
> >
> > ---
> >
> > **Q5: Editorial comments**
> >
> > **A5.** Thank you for your suggestion. We will incorporate them in the final draft by including the optimization objective for MWM in the main draft.

---

> > > ### Comment · Reviewer_GUgv · 2022-08-27
> > > **Thank you for your response**
> > >
> > > Thank you for conducting the additional experiments. I look forward to seeing the manuscript with all of them.

---

> > > > ### Author Response · Authors · 2022-08-27
> > > > **Response**
> > > >
> > > > Thank you for your response, we will incorporate additional experimental results into the final draft.
> > > >
> > > > Thank you again for your valuable comments!
> > > >
> > > > Best,
> > > >
> > > > Authors

---

### Official Review · Reviewer_abdd · 2022-07-31

**Originality:** Good
**Technical Quality:** Fair
**Clarity Of Presentation:** Good
**Impact:** 3

**Recommendation:**

Weak Accept: I recommend accepting the paper, but will not argue for my recommendation if the majority of other reviewers have a different opinion.

**Summary:**

The paper considers model-based RL with visual input. The key idea is to decouple visual representation learning and dynamics learning. For visual representation learning, inspired by the recent masked auto-encoder (MAE), the paper trains an autoencoder with convolutional layers and vision transformers (ViT) to reconstruct pixels given masked convolutional features. To encode task-specific information, the paper introduces an auxiliary reward prediction objective for the autoencoder. For dynamics learning, the paper adopts the DreamerV2 approach to learn a latent dynamics model that operates on visual representations from the autoencoder. The paper presents performance results on a variety of visual robotic tasks from Meta-world and RLBench.

**Issues:**

It seems that predicting reward is very important to achieve high performance. The paper should provide more information on what features  are enabled by predicting reward. How does this finding change with different environment and tasks? Since the paper evaluates only in a simulator, can reward be effectively predicted for real world data?

It seems that conv feature masking is a type of dropout. Instead of conv feature masking, if dropout is used, will it achieve the same performance results?

**Quality Of The Limitations Section:**

Limitations are not well addressed

**Reviewer Expertise:**

4: The reviewer is confident but not absolutely certain that the evaluation is correct

**Robotics Focus:**

Highly relevant to robotics but no hardware experiments

**Strengths And Weaknesses:**

Strengths

The paper proposes MWM consisting of an autoencoder with convolutional layers and vision transformers (ViT) to reconstruct pixels given masked convolutional features,  an auxiliary reward prediction objective for the autoencoder, a latent dynamics model that operates on visual representations from the autoencoder.

It shows that self-supervised ViT trained to reconstruct visual observations with convolutional feature masking can be effective for visual model-based RL.

Weaknesses

How MWM differs from DreamerV2 is not clearly stated. The main difference seems to be a redesign of the representation model and image decoder. The two models of MWM use ViT encoder and decoder with conv feature masking.

There are several questions regarding the comparison with DreamerV2. What codebase does the paper use? Since DreamerV2 only evaluated on Atari games, does the paper make any changes to DreamerV2? The poor performance might be due to no adaptation of model architecture and hyperparameters. How does MWM compare with DreamerV2 on Atari games?

The most relevant paper is cited in appendix,
T. Xiao, I. Radosavovic, T. Darrell, and J. Malik. Masked visual pre-training for motor control. arXiv preprint arXiv:2203.06173, 2022.

which uses features from a masked auto-encoder. The paper should compare MWM with this paper.





**Summary Of Recommendation:**

The paper introduces masked auto-encoder features and conv feature masking to DreamerV2. This redesign of DreamerV2 inspired by vision transformer and masked auto-encoder seems to be very appealing and promising since the feature learning can potentially be much better. However, the comparison with DreamerV2 lacks details. Comparing DreamerV2 designed for Atari games on Meta-world and RLBench with no change does not seem to be convincing. One way to address this concern is to compare MWM directly with DreamerV2 on Atari games.

Furthermore, the Xiao et al. paper has applied masked auto-encoder to motion control. The paper should also consider comparing with the this most related paper cited in appendix.

===
Several of my concerns are clarified during rebuttal. I upgrade my rating to weak accept from weak reject.

---

> ### Author Response · Authors · 2022-08-24
> **Response to Reviewer abdd (1/2)**
>
> Dear reviewer abdd,
>
> We sincerely appreciate your valuable comments. We found them extremely helpful in improving our draft. We address each comment in detail, one by one below.
>
> ---
>
> **Q1: Clarification on the difference to DreamerV2 architecture**
>
> **A1.** Thanks for pointing out the missing details. The main difference between MWM and DreamerV2 is in the representation model and the image decoder, as you mentioned. In more detail:
> The representation model of DreamerV2 takes CNN embeddings as inputs, and the dynamics model is trained using the learning signal from reconstructing raw image observations using the CNN decoder.
> On the other hand, The representation model of MWM takes representations from the autoencoder, aggregated with a 2-layer ViT encoder, as inputs. Then the dynamics model is trained using the learning signal from reconstructing autoencoder representations using the 2-layer ViT decoder.
> Here, an important difference is that the CNN encoder in DreamerV2 is optimized with the gradients from the dynamics model learning, while MWM learns autoencoder representations separately. We will further clarify this in the final draft.
>
> ---
>
> **Q2: Codebase**
>
> **A2.** We use the [official DreamerV2 codebase](https://github.com/danijar/dreamerv2). We have also attached our source code for MWM in the supplementary material.
>
> ---
>
> **Q3: Concern on the performance of DreamerV2**
>
> **A3.**  We would like to clarify that DreamerV2 is generally applicable to various control tasks, including continuous control tasks, e.g., DeepMind Control Suite, and not designed specifically for Atari games. For instance, DreamerV2 official codebase support experimentation on DeepMind Control Suite tasks with a set of default hyperparameters, which achieves overall better performance than Dreamer (raw data is available at [here](https://github.com/danijar/dreamerv2/blob/main/scores/dmc-vision-dreamerv2.json)). Our experimental results are based on these hyperparameters that are further tuned to make DreamerV2 achieve non-zero success rates on most of the Meta-world tasks. Specifically, following a prior work [1] that reported the performance of DreamerV2 on Meta-world, we use
> - Larger batch size (16 -> 50)
> - Larger recurrent network (200 units -> 1024 units)
> - Reward normalization crucial for achieving consistent performance across 50 tasks.
>
> We will clarify these experimental details in the final draft.
>
> We also note that our DreamerV2 results are much stronger on continuous control tasks than the official state-of-the-art results due to updated hyperparameters. For instance, DreamerV2 with our hyperparameters achieves ~800 episode return on Quadruped Walk with 0.5 environment steps (Figure 5 of the main draft), while DreamerV2 with original hyperparameters achieves ~400. This implies that the weak performance of DreamerV2 on manipulation tasks is not from model architecture or hyperparameters tuned for Atari games with discrete action spaces. So, we believe that our experimental results with carefully tuned DreamerV2 agent are convincing and clearly show that our approach of decoupling representation and dynamics learning is indeed effective for visual control tasks.
>
> [1] Seo, Younggyo, Kimin Lee, Stephen L. James, and Pieter Abbeel. "Reinforcement learning with action-free pre-training from videos." In International Conference on Machine Learning (ICML), 2022.
>
> ---
>
> **Q4: Comparison with MVP**
>
> **A4.** We would like to clarify that MVP [2] is concurrent and orthogonal research to MWM in that it focuses on investigating the effectiveness of pre-training on real-world videos, and we believe investigating how such a real-world pre-training can be effective for MWM would be a very interesting future direction.
>
> [2] Xiao, Tete, Ilija Radosavovic, Trevor Darrell, and Jitendra Malik. "Masked visual pre-training for motor control." arXiv preprint, 2022.

---

> > ### Author Response · Authors · 2022-08-24
> > **Response to Reviewer abdd (2/2)**
> >
> > **Q5: Additional analysis on reward prediction**
> >
> > **A5.** Thanks for your suggestion to provide more information on how the reward prediction enables the model to achieve better performance. Our hypothesis is that reward prediction can provide information useful for robotic manipulation, which can also be shared across various tasks. For instance, rewards are often designed to contain information about robot arm movements (translations, rotations, and grasping) and fine-grained details about objects, which can be shared across various manipulation tasks.
> >
> > To empirically investigate this hypothesis, we first train MWM on Meta-world Push task with and without reward prediction, and train a regression model to predict the states in Push (seen) and Pick-place (unseen) tasks on top of frozen autoencoder representations from Push task. In our experiments (https://imgur.com/a/2rnTBje), we observe that representations trained with reward prediction consistently achieve low prediction error. Moreover, we also report the performance of MWM on Pick-place with these frozen representations, where we find that reward prediction enables generalization across significantly different tasks. This supports that reward prediction can indeed provide useful information for robotic manipulation.
> >
> > ---
> >
> > **Q6: Can reward be predicted for real-world data?**
> >
> > **A6.** Given that recent work [3] has demonstrated that world models with reward prediction can work well for training real robots from camera observations with both dense and sparse rewards, we expect MWM also has the potential to work well for real-world scenarios.
> >
> > [3] Wu, Philipp, Alejandro Escontrela, Danijar Hafner, Ken Goldberg, and Pieter Abbeel. "DayDreamer: World Models for Physical Robot Learning." arXiv preprint, 2022.
> >
> > ---
> >
> > **Q7: Would dropout achieve the performance as convolutional feature masking?**
> >
> > **A7.** This is an interesting question. As you mentioned, convolutional feature masking can be seen as applying dropout to features from early convolution layers, so dropout might achieve a similar performance to our approach. However, it would incur quadratically more compute costs than the current architecture. This is because all the features would be given as inputs to the ViT encoder when we use dropout, unlike our model that only takes un-masked features as inputs following the design choice of MAE.

---

> > > ### Comment · Reviewer_abdd · 2022-08-26
> > > **Thank you**
> > >
> > > The authors have clarified several of my concerns. I upgrade my rating to weak accept.

---

> > > > ### Author Response · Authors · 2022-08-26
> > > > **Thank you for the response**
> > > >
> > > > We are glad to hear that our response addressed your concerns well.
> > > >
> > > > Thank you again for your valuable comments!
> > > >
> > > > Best,
> > > >
> > > > Authors

---

### Official Review · Reviewer_DGEG · 2022-08-06

**Originality:** Good
**Technical Quality:** Very Good
**Clarity Of Presentation:** Very Good
**Impact:** 4

**Recommendation:**

Weak Accept: I recommend accepting the paper, but will not argue for my recommendation if the majority of other reviewers have a different opinion.

**Summary:**

This paper presents Masked World Models (MVW), a model-based reinforcement learning framework for visual control.

There are a few unique designs of MVM compared to existing approaches:
- MVM decouples visual representation learning from dynamics learning, this is different from the existing Dreamer framework that uses a single model to learn both visual representations and dynamics together.
- MVM uses a self-supervised Visual Transformer for representation learning, and it performs masking on convolutional features, instead of masking on pixels.
- MVM also uses an auxiliary reward prediction to encode task-relevant information.

Experiments on synthetic datasets show that MVW significantly outperforms the state of the art.

**Issues:**

I am curious does decoupling the visual learning from the dynamics learning make it "slower" to train the approach, compared to having a single network doing both tasks?

**Quality Of The Limitations Section:**

Limitations are addressed clearly

**Reviewer Expertise:**

3: The reviewer is fairly confident that the evaluation is correct

**Robotics Focus:**

Highly relevant to robotics but no hardware experiments

**Strengths And Weaknesses:**

Strengths:
- It seems to me that, MVM incorporates a few design changes (mentioned in the summary part of my review) to the Dreamer approach and showed that these design changes lead to significant performance gain.
- The paper reads very well. Although I am not in this field, I was able to quickly grasp the related work and the unique contributions of this paper.

Weaknesses:
- Although the paper is well written, the contributions seem incremental -- it is adding small changes and tweaks to existing approaches.
- Real-world robotic experiments would be more convincing.

**Summary Of Recommendation:**

Overall, interesting ideas and good results. I vote for weak accept.

---

> ### Author Response · Authors · 2022-08-24
> **Response to Reviewer DGEG**
>
> Dear reviewer DGEG,
>
> We sincerely appreciate your valuable comments. We found them extremely helpful in improving our draft. We address each comment in detail, one by one below.
>
> ---
>
> **Q1: Contribution**
>
> **A1.** We emphasize that our key contribution is first to demonstrate that visual model-based RL can work on more challenging robotic domains such as Meta-world and RLBench by decoupling visual representation and dynamics learning. We also have demonstrated that two design choices of our autoencoder (i.e., early convolution and reward prediction) are crucial to enable the decoupling approach to be effective, with extensive ablation studies to support the claims. Moreover, while each component (MAE, early convolutions, reward prediction) is already known, we would like to emphasize that the combination of these component for better representation learning with an eye toward visual control tasks, e.g., capturing fine-grained details, is novel. So, we believe our contributions are significant and bring novel insights to the robot learning community. We will further clarify and emphasize our contributions in the final draft.
>
> ---
>
> **Q2: “I am curious does decoupling the visual learning from the dynamics learning make it "slower" to train the approach, compared to having a single network doing both tasks?”**
>
> **A2.** Our finding is that decoupled approach can actually be faster in learning to solve visual control tasks because each model can specialize in each task. Specifically, visual encoders can be quickly trained using fewer samples via advanced self-supervised learning techniques on images, and the dynamics model can solely focus on learning dynamics information using better representations from the beginning of the training. On the other hand, a single model for both tasks would suffer from a tradeoff between two tasks due to the limited capacity of the model. We would also like to mention that this is also aligned with the recent trend in video generation where a two-stage approach of separately training image generators (i.e., VQ-VAE, VQ-GAN, MAE) and autoregressive Transformer [1,2,3,4]. Thanks for your interesting question, and we will include the relevant discussion in the final draft.
>
> [1] Yan, Wilson, Yunzhi Zhang, Pieter Abbeel, and Aravind Srinivas. "Videogpt: Video generation using vq-vae and transformers." arXiv preprint, 2021.
>
> [2] Rakhimov, Ruslan, Denis Volkhonskiy, Alexey Artemov, Denis Zorin, and Evgeny Burnaev. "Latent video transformer." in Proceedings of the 16th International Joint Conference on Computer Vision, Imaging and Computer (VISIGRAPP), 2021.
>
> [3] Wu, Chenfei, Jian Liang, Lei Ji, Fan Yang, Yuejian Fang, Daxin Jiang, and Nan Duan. "N\" uwa: Visual synthesis pre-training for neural visual world creation." arXiv preprint, 2021.
>
> [4] Gupta, Agrim, Stephen Tian, Yunzhi Zhang, Jiajun Wu, Roberto Martín-Martín, and Li Fei-Fei. "Maskvit: Masked visual pre-training for video prediction." arXiv preprint, 2022.
>
> ---
>
> **Q3: Lack of hardware experiments**
>
> **A3.** We agree with the reviewer that missing hardware experiments is one of our weaknesses, and including them could have further strengthened the contribution of our paper. Given that recent work [5] has demonstrated that world models can be used for training real robots from camera observations, experimentation with MWM in real robots would be an interesting investigation.
>
> [5] Wu, Philipp, Alejandro Escontrela, Danijar Hafner, Ken Goldberg, and Pieter Abbeel. "DayDreamer: World Models for Physical Robot Learning." arXiv preprint, 2022.

---

### Official Review · Reviewer_iNVH · 2022-08-07

**Originality:** Good
**Technical Quality:** Good
**Clarity Of Presentation:** Very Good
**Impact:** 3

**Recommendation:**

Weak Accept: I recommend accepting the paper, but will not argue for my recommendation if the majority of other reviewers have a different opinion.

**Summary:**

The paper presents a method for decoupling visual representation and latent dynamics learning for control. It does so by combining  recent approaches for visual representation learning (ViT) and latent dynamics learning (Dreamer) for control such that they are decoupled. This decoupling allows for learning a better prediction model as demonstrated through experiments in a diverse set of environments.

**Issues:**

- Missing relevant works specifically with respect to decoupled visual representation learning and dynamics learning:
	- [1] uses an special bottlenecked autoencoder that generates relevant 2D keypoints on the image for learning visual representation separately from learning a dynamics model on top of these features.  [2, 3] demonstrate this method's effectiveness on hardware including arm robots [3]. This method can be considered an additional baseline.
	- Another relevant work in this regard includes  [4] which learns Dense Descriptions as visual representation  and learns a dynamics model from these representations separately. This method requires RGBD images, however.
- Minor:
	- On Page 3, CLS is not defined.

References :

[1] Minderer, M., Sun, C., Villegas, R., Cole, F., Murphy, K. P., & Lee, H. (2019). Unsupervised learning of object structure and dynamics from videos. Advances in Neural Information Processing Systems, 32.

[2] Lambeta, M., Chou, P. W., Tian, S., Yang, B., Maloon, B., Most, V. R., ... & Calandra, R. (2020). Digit: A novel design for a low-cost compact high-resolution tactile sensor with application to in-hand manipulation. IEEE Robotics and Automation Letters, 5(3), 3838-3845.

[3] Das, N., Bechtle, S., Davchev, T., Jayaraman, D., Rai, A., & Meier, F. (2020). Model-based inverse reinforcement learning from visual demonstrations. CoRL 2020.

[4] Manuelli, L., Li, Y., Florence, P., & Tedrake, R. (2020). Keypoints into the future: Self-supervised correspondence in model-based reinforcement learning. arXiv preprint arXiv:2009.05085.

**Quality Of The Limitations Section:**

Limitations are addressed clearly

**Reviewer Expertise:**

3: The reviewer is fairly confident that the evaluation is correct

**Robotics Focus:**

Highly relevant to robotics but no hardware experiments

**Strengths And Weaknesses:**

Strengths:
- The paper is well written and well structured
- has a number of interesting experiments that show the proposed methods superiority to the chosen baseline Dreamer V2 as well as an extensive ablation study
- Learning the reward alongside latent features in ViT is an interesting way of encoding task relevant information in visual representation architecture.

Weaknesses
- Missing relevant works
- Only one baseline that does not decouple visual representation and dynamics learning
- Will the auxilliary reward prediction in the visual representation learning not limit the generalizability of the learned visual representor to only one RL task?
- Minor issues such as missing acronym expansion
- Lack of hardware experiments

Questions/Doubts:
- In Figure 6.a, is the plot for Pixel patch (MAE) produced with reward learning?

**Summary Of Recommendation:**

The paper combines two recent approaches for visual representation learning and latent dynamics learning for control such that they are decoupled. While this proposal is not entirely novel, this paper demonstrates the advantage of this decoupling empirically with several experiments in different simulation environments. There is one novel idea (in my opinion) - the auxilliary reward learning for encoding task relevant information. However, this might limit the learned visual representation's generalizability to different tasks.

---

> ### Author Response · Authors · 2022-08-24
> **Response to Reviewer iNVH**
>
> Dear reviewer iNVH,
>
> We sincerely appreciate your valuable comments. We found them extremely helpful in improving our draft. We address each comment in detail, one by one below.
>
> ---
>
> **Q1: Additional decoupled baseline**
>
> **A1.** To address your concern, we additionally report the performance of a baseline that decouples visual representation and dynamics by learning VAE with an auxiliary reward prediction task. In our experiments (https://imgur.com/a/XXaAYs6), we find that the baseline (i.e., Decoupled w/ VAE) fails to solve Reach, which is one of the easiest tasks in Meta-world, even though the reconstructions are almost perfect. This shows that representation learning with our autoencoder is crucial for performance improvement from the decoupling approach. We will try to include results with more baselines in the final draft, also including keypoint-based baseline [1] you suggested.
>
> [1] Minderer, Matthias, Chen Sun, Ruben Villegas, Forrester Cole, Kevin P. Murphy, and Honglak Lee. "Unsupervised learning of object structure and dynamics from videos." Advances in Neural Information Processing Systems (NeurIPS), 2019.
>
> ---
>
> **Q2: Generalizability of visual representations learned with reward prediction**
>
> **A2.** We would like to remark that reward prediction on a specific task can also encourage visual representations to capture task-irrelevant information useful for solving various manipulation tasks. For instance, rewards are often designed to contain information about robot arm movements (translations and rotations) and fine-grained details about objects, which can be shared across various manipulation tasks.
>
> To empirically support this, we provide additional experimental results (https://imgur.com/a/IHuRxhZ) where we utilize frozen representations trained on push task for solving manipulation tasks with a difference to Push task: (i) Push Back that requires moving block to a different direction, (ii) Pick Place that requires picking up the block, and (iii) Drawer Open that contains unseen drawer object in the observation. In our experiments, we observe that performance with frozen representations can be similar to or better than the performance of MWM trained from scratch, which shows that representations learned with reward prediction can be versatile.
>
> ---
>
> **Q3: “In Figure 6.a, is the plot for Pixel patch (MAE) produced with reward learning?”**
>
> **A3.** Thanks for pointing out the missing details. Yes, Pixel patch (MAE) in Figure 6(a) is trained with reward prediction. We will clarify this in the final draft.
>
> ---
>
> **Q4: Lack of hardware experiments**
>
> **A4.** We agree with the reviewer that missing hardware experiments is one of our weaknesses, and including them could have further strengthened the contribution of our paper. Given that recent work [2] has demonstrated that world models can be used for training real robots from camera observations, experimentation with MWM in real robots would be an interesting investigation.
>
> [2] Wu, Philipp, Alejandro Escontrela, Danijar Hafner, Ken Goldberg, and Pieter Abbeel. "DayDreamer: World Models for Physical Robot Learning." arXiv preprint, 2022.
>
> ---
>
> **Q5: Missing relevant works & Editorial comments**
>
> **A5.** Thanks for the pointer to missing related works and editorial comments. We will incorporate them into the final draft.

---

> > ### Comment · Reviewer_iNVH · 2022-08-26
> > **Response**
> >
> > Thank you for the additional experiments - especially the ones regarding the generalizability of representations learned with reward prediction! However, I didn't find the decoupling results from https://imgur.com/a/XXaAYs6 very convincing. I still think that there are stronger decoupling baselines (see my comment regarding relevant works) that your approach should be compared against rather than simply VAE with reward prediction - you already show that your approach does well against (decoupled?) MAE (which I expect fares better or equal to VAE in terms of reconstruction), so it is not surprising that VAE woudn't do so well either.
> >
> > I will still keep my weak accept rating (mainly since I found auxillary reward prediction to be an interesting addition). However I will urge the authors to also provide results with stronger baselines.

---

> > > ### Author Response · Authors · 2022-08-26
> > > **Thank you for your response**
> > >
> > > We are glad to hear that you liked our response about the generalizability of reward prediction.
> > >
> > > While we could not provide the results during the rebuttal due to time and compute constraints, following your suggestion, we will try our best to provide results with stronger baselines in the final draft.
> > >
> > > Thank you again for your valuable comments!
> > >
> > > Best,
> > >
> > > Authors

---

### Official Review · Reviewer_btzk · 2022-08-08

**Originality:** Fair
**Technical Quality:** Fair
**Clarity Of Presentation:** Good
**Impact:** 3

**Recommendation:**

Weak Reject: I recommend rejecting the paper, but will not argue for my recommendation if the majority of other reviewers have a different opinion.

**Summary:**

This paper improves the DreamerV2 model-based RL framework, by replacing the the discrete world model encoder (looks like a VQ-VAE style encoder with recurrency? but I' m not sure) to a MAE-like encoder (The main difference is that it's masking the feature instead of the origin image). Experiments show that the proposed method achieves much better performance tha the original DreamerV2.

**Issues:**

- Concerns about experiment details mentioned in the Weakness section.
- There are some obvious mistakes in the figures. For example, I noticed that unit of success rate is "%" yet the scale is from 0.0 - 1.0. Obviously the author means 0.0 - 100.0.

**Quality Of The Limitations Section:**

Additional details required

**Reviewer Expertise:**

2: The reviewer is willing to defend the evaluation, but it is quite likely that the reviewer did not understand central parts of the paper

**Robotics Focus:**

Relevant but unlikely to deploy to hardware in near future

**Strengths And Weaknesses:**

- Strength: It achieves much better result than the baseline method in several tasks in the settings that the author propose.

- Weakness: Experiment settings seem to have a huge gap between this submission and the original DreamerV2 paper. To be more specific, in the dreamer V2 paper (https://arxiv.org/pdf/2010.02193.pdf), it reports result after 4e7 steps for Humanoid Walk (2010.02193.pdf Appendix Fig A.2). Yet in this paper, there seems to be no more than 3e6 steps executed in any of its setups. As we all know, the training process of RL agents are unstable and the may suffer from zero-reward at the beginning of its training process, it would be important to wait until the training process to converge. Furthermore, in Figure 3 of this submission, Hand Insert / Shelf Place task increase concern about this issue, where we notice the DreamerV2 reward begin to increase after ~2e6/~3e6 steps, which is quite similar to 2010.02193.pdf Appendix Fig A.2.

I'm also quite confused why the paper does not report any Atari Game or Humanoid task in the DM Control Suite, which are reported by the original DreamerV2 paper.

Since the core contribution of the submission is additional (i.e. change the encoder), experiments details and solidity are important.

**Summary Of Recommendation:**

See the Weakness section. My main concern about this paper is the experiment part that, as has been mentioned in the Weakness section. Since the core contribution of the submission is additional (i.e. change the encoder), experiments details and solidity are important. Since the experiment setups may be not aligned with the DreamerV2 paper, and the paper does not report any Atari Game or Humanoid task in the DM Control Suite, which are reported by the original DreamerV2 paper, I am not very certain if the experiment part of this paper is convincing enough.

That being said, I myself do not have actual experience training model-based RL agents, so I rate the confidence (Reviewer Expertise) to be 2. If there are any severly technically wrong part of my review please the authors and the ACs point them out.

---

> ### Author Response · Authors · 2022-08-24
> **Response to Reviewer btzk**
>
> Dear reviewer btzk,
>
> We sincerely appreciate your valuable comments. We found them extremely helpful in improving our draft. We address each comment in detail, one by one below.
>
> ---
>
> **Q1: Contribution**
>
> **A1.** We emphasize that our key contribution is not just to replace the encoder but first to demonstrate that visual model-based RL can work on more challenging robotic domains such as Meta-world and RLBench by decoupling visual representation and dynamics learning. We also have demonstrated that two design choices of our autoencoder (i.e., early convolution and reward prediction) are crucial to enable the decoupling approach to be effective, with extensive ablation studies to support the claims. So, we believe our contributions are significant and bring novel insights to the robot learning community. We will further clarify and emphasize our contributions in the final draft.
>
> ---
>
> **Q2: Concern on experimental setup due to short environment steps**
>
> **A2.** We agree that it would be insightful to investigate the effectiveness of our approach with more steps, and we will include the corresponding results in the final draft due to time and compute constraints from longer training. Here, we would also like to emphasize that we did not report the performance over longer steps because our main focus is sample-efficiency. We are using simulator as a proxy to eventually being able to train real-robots, where it is impractical to train robots for 1e7 steps. We will further clarify this in the final draft.
>
> ---
>
> **Q3: No Atari and Humanoid experiments**
>
> **A3.** We focus on robot learning, especially continuous control for robotic manipulation tasks, which requires the ability to recognize small objects and interact with them. Because of that, we did not consider Atari games (discrete control, without robot learning) and Humanoid (without small objects). Nonetheless, following your suggestion, we will also report the performance of MWM on Humanoid tasks. We will include the corresponding results in the final draft due to time and compute constraints of longer training required for solving Humanoid tasks.
>
> ---
>
> **Q4: Results are not convincing**
>
> **A4.** We emphasize that we tuned default hyperparameters for continuous control with DreamerV2 to make it achieve non-zero success rates on most of the Meta-world tasks. Specifically, following a prior work [1] that reported the performance of DreamerV2 on Meta-world, we use
> - Larger batch size (16 -> 50)
> - Larger recurrent network (200 units -> 1024 units)
> - Reward normalization crucial for achieving consistent performance across 50 tasks.
>
> We will clarify these experimental details in the final draft.
>
> We also note that our DreamerV2 results are much stronger on continuous control tasks than the official state-of-the-art results (raw data is available at [here](https://github.com/danijar/dreamerv2/blob/main/scores/dmc-vision-dreamerv2.json)) due to updated hyperparameters. For instance, DreamerV2 with our hyperparameters achieves ~800 episode return on Quadruped Walk with 0.5 environment steps (Figure 5 of the main draft), while DreamerV2 with original hyperparameters achieves ~400. So, we believe that our experimental results with carefully tuned DreamerV2 agent are convincing and clearly show that our approach of decoupling representation and dynamics learning is indeed effective for visual control tasks.
>
> [1] Seo, Younggyo, Kimin Lee, Stephen L. James, and Pieter Abbeel. "Reinforcement learning with action-free pre-training from videos." In International Conference on Machine Learning (ICML), 2022.
>
> ---
>
> **Q5: Editorial comments**
>
> **A5.** Thanks for pointing this out! We will modify the scale of success rates in the final draft.

---

### Meta-Review · Area_Chair_eJCY · 2022-08-07

**Recommendation:** Accept (Poster)
**Confidence:** 4

**Metareview:**

This paper introduces an model-based RL approach that decouples visual representation learning and dynamics learning using a ViT. Results show that both this decoupling and the replacement encoder result in improved performance on a range of simulated benchmarking suites and tasks (RLBench, metaworld, dm_control). In the initial review process reviewers noted that the paper was

* Well written, interesting idea and a unique model design
* Good performance demonstrated in simulation and well structured ablations

but expressed concerns about
* Some missing literature, and questions about the strength/ suitability of the baselines (eg. no comparison with other decoupled models, baseline tuning)
* No experiments on real hardware
* Potential limitations of the approach re versatility due to the inclusion of reward in the representation learning task, which learns task conditioned dynamics models

The authors rebuttal included some additional baselines, argued that there is evidence that similar models do transfer to real world settings, and added additional results to address concerns about overfitting to tasks by conditioning on reward (reward signal can be task agnostic or aid in representation learning). Reviewers were generally positive about this work post rebuttal, so I am recommending acceptance. Although there is a strong focus on real robot experiments at this years CoRL, I believe the extensive number of experiments on multiple benchmarks makes up for the missing real robot experiments to some extent.

I encourage the authors to include the additional experiments in the supplementary material included with the camera ready paper, along with additional clarifications and discussion points raised during the review process.

---

> ### Author Response · Authors · 2022-08-28
> **Response to Area Chair eJCY**
>
> Dear Area Chair,
>
> We sincerely appreciate your time and effort in reviewing our draft. We would like to provide the summary of our responses and discussion with the reviewers about weaknesses in the meta-review as below:
>
> ---
>
> **Q1. Missing literature (Reviewer iNVH)**
>
> **A1.** As we have mentioned in our response **A5** to Reviewer iNVH, we will include the missing related works pointed out by the reviewer in the final draft.
>
> ---
>
> **Q2. Strength of the baseline (Reviewer abdd, btzk)**
>
> **A2.** During the discussion phase, we are glad to find that our response **A3** to Reviewer abdd successfully addressed the concern on this. Specifically, We have clarified that our DreamerV2 baseline is a valid and strong baseline for continuous control tasks with additionally tuned hyperparameters that further improve performance. We believe that our response **A4** to Reviewer btzk can also address the concern of the reviewer, who has not yet responded at the time of preparing for this response.
>
> ---
>
> **Q3. Additional baseline that decouples visual representation and dynamics learning (Reviewer iNVH)**
>
> **A3.** In our response **A1** to Reviewer iNVH, we have provided the performance of an additional decoupled baseline that separately learns VAE with reward prediction, where we find that MWM significantly outperforms this baseline. In the final draft, we will try our best to provide stronger baselines following the suggestion of Reviewer iNVH.
>
> ---
>
> **Q4. Generalizability of representations trained with reward prediction (Reviewer abdd, iNVH, GUgv)**
>
> **A4.** During the discussion phase, we find that our response successfully addressed the concern of all reviewers (Reviewer abdd, iNVH, GUgv). In particular, we are glad that Reviewer iNVH highlighted that our response on reward prediction is very convincing. Specifically, we explained that reward prediction on a specific task can also encourage visual representations to capture task-irrelevant information useful for solving various manipulation tasks. To support this, we have provided additional experimental results (https://imgur.com/a/IHuRxhZ) where we utilize frozen representations trained on push task for solving manipulation tasks with a difference to push task: (i) Push Back, (ii) Pick Place, and (iii) Drawer Open. We observe that performance with frozen representations can be similar to or better than the performance of MWM trained from scratch, which shows that representations learned with reward prediction can be versatile.
>
> ---
>
> **Q5. Lack of hardware experiments (Common)**
>
> **A5.** We agree with the reviewers that missing hardware experiments is one of our weaknesses, and including them could have further strengthened the contribution of our paper. Given that recent work [1] has demonstrated that world models can be used for training real robots from camera observations, experimentation with MWM in real robots would be an interesting investigation.
>
> [1] Wu, Philipp, Alejandro Escontrela, Danijar Hafner, Ken Goldberg, and Pieter Abbeel. "DayDreamer: World Models for Physical Robot Learning." arXiv preprint, 2022.
>
> ---
>
> With valuable feedback from the reviewers and discussion with them during this phase, we believe that we have successfully addressed all the points raised by the reviewers and that these clarifications and additional results will further strengthen our paper.
>
> Thank you very much.
>
> Authors